# Towards deeper understanding of multifaceted chemistry of magnesium alkylperoxides

Tomasz Pietrzak [1], Iwona Justyniak[2], Karolina Zelga[1], Krzysztof Nowak [1], Zbigniew Ochal[1] & Janusz Lewiński [1,2✉]

Despite considerable progress in the multifaceted chemistry of non-redox-metal alkylperoxides, the knowledge about magnesium alkylperoxides is in its infancy and only started to gain momentum. Harnessing the well-defined dimeric magnesium tert-butylperoxide [($^{f5}$BDI)Mg($\mu$-$\eta^2$:$\eta^1$-OOtBu)]$_2$ incorporating a fluorinated $\beta$-diketiminate ligand, herein, we demonstrate its transformation at ambient temperature to a spiro-type, tetranuclear magnesium alkylperoxide [($^{f5}$BDI)$_2$Mg$_4$($\mu$-OOtBu)$_6$]. The latter compound was characterized by single-crystal X-ray diffraction and its molecular structure can formally be considered as a homoleptic magnesium tert-butylperoxide [Mg($\mu$-OOtBu)$_2$]$_2$ terminated by two monomeric magnesium tert-butylperoxides. The formation of the tetranuclear magnesium alkylperoxide not only contradicts the notion of the high instability of magnesium alkylperoxides, but also highlights that there is much to be clarified with respect to the solution behaviour of these species. Finally, we probed the reactivity of the dimeric alkylperoxide in model oxygen transfer reactions like the commonly invoked metathesis reaction with the parent alkylmagnesium and the catalytic epoxidation of trans-chalcone with tert-butylhydroperoxide as an oxidant. The results showed that the investigated system is among the most active known catalysts for the epoxidation of enones.

[1] Faculty of Chemistry, Warsaw University of Technology, Warsaw, Poland. [2] Institute of Physical Chemistry, Polish Academy of Sciences, Warsaw, Poland. ✉email: lewin@ch.pw.edu.pl

Non-redox-metal alkylperoxides ($^{NR}$MOOR) represent an important class of reactive intermediates that may act as efficient oxidants in oxygen-transfer organic transformations like the epoxidation of enones[1–3] or synthesis of α-aminoperoxides[4]. Moreover, a combination of non-redox-metal-based organometallic complexes with dioxygen have also been exploited as radical initiators for various organic reactions[5–8]. While the formation of $^{NR}$MOOR species was already been postulated in 1890[9], knowledge on their multifaceted chemistry is rather scarce, which essentially hamper the further development of stoichiometric and catalytic systems involving $^{NR}$MOOR complexes. Owing to frequently encountered exceptional reactivity of $^{NR}$MOOR species, the controlled synthesis and subsequent isolation of this family of compounds still appear to be a challenge[9–11]. In the last three decades, a few examples of heteroleptic Group 1[12], 2[13–15], 12[16–21] and 13[22–36] metal alkylperoxides have occasionally been isolated. In particular, systematic research on the oxygenation chemistry of organozincs have marked an important milestones in a deeper understanding of the reactivity of ZnOOR species. For example, it has been convincingly proved that the O–O bond scissions in an alkylperoxide moiety, affording the oxyl ZnO• and alkoxy •OR radicals, precedes for the formation of variety of products spanning from zinc oxo[37,38] and hydroxide[39–41] even to carboxylate[42,43] clusters. Our recent studies also strongly contradicted to the widely accepted textbook mechanism involving the metathesis reaction between a highly reactive $^{NR}$MOOR and the parent $^{NR}$MR moieties[16,43]. In turn, homoleptic compounds of the type $[^{NR}MOOR]_n$ are scant and only limited to structurally characterized lithium dodecameric clusters $[LiOOR]_{12}$ (R = $t$Bu[44], CMe$_2$Ph[12]) and a dimeric $[(LiOOtBu)(tBuOOH)]_2$ compound[25]. All these lithium alkylperoxides were synthesized by the protonolysis of the parent lithium precursors with the respective organic hydroperoxide. To best of our knowledge, no structure has yet been reported for a homoleptic magnesium alkylperoxide.

More importantly, the knowledge about magnesium alkylperoxides is in its infancy and only started to gain momentum[13–15,45,46]. Although the first records about the putative formation of MgOOR species came from 1909[9], it was not until 2003 that the first magnesium alkylperoxide complex was structurally characterized[13]. However, this magnesium alkylperoxide stabilized by a β-diketiminate ligand was isolated as co-crystals with the corresponding magnesium alkoxide complex which raised doubts as to the reactivity and stability of the magnesium alkylperoxide species. Strikingly, in 2016 our group reinvestigated this reaction system and showed that the selective formation of magnesium alkylperoxide is viable in a highly controlled manner, whilst the formation of the magnesium alkoxide is due to the transformation of the alkylperoxide complex at room temperature (Fig. 1a)[14]. Recently, the first monomeric magnesium alkylperoxide was successfully isolated via the controlled oxygenation of neo-pentylmagnesium complex with the β-diketimine ligand[15].

To investigate the influence of steric and electronic contributions of a supporting ligand on the reactivity of magnesium alkylperoxides, we turned our attention to β-diketimine ligands with fluorinated N-aryl substituents. Previously, we demonstrated that the oxygenation of a $tert$-butylmagnesium complex incorporating deprotonated 2-[(2,4,6-trifluorophenyl)amino]−4[(2,4,6-trifluorophenyl)imino]-pent-2-ene (hereafter $^{f3}$BDI) led to the respective alkylperoxide or alkoxide compounds, depending on the reaction temperature (Fig. 1a). Moreover, introduction of fluorine atoms into the N-aryl substituents of the β-diketimine ligand has major effect on the catalytic activity of the resulting alkylperoxide $[(^{f3}BDI)Mg(\mu-OOtBu)]_2$ in the epoxidation of electron-deficient olefins in comparison to the non-fluorinated $[(^{dipp}BDI)Mg(\mu-OOtBu)]_2$-based catalytic system[14]. Advancing the chemistry of magnesium alkylperoxides, now we turn our attention to deprotonated 2-[(2,3,4,5,6-pentafluorophenyl)amino]-4[(2,3,4,5,6-pentafluorophenyl)imino]-pent-2-ene (hereafter $^{f5}$BDI) as a β-diketiminate ligand with perfluorinated N-aryl substituents. Herein, we report the transformation of a new dimeric magnesium $tert$-butylperoxide $[(^{f5}BDI)Mg(\mu-\eta:^2\eta^1-OOtBu)]_2$ complex to a spiro-type, tetranuclear magnesium alkylperoxide $[(^{f5}BDI)_2Mg_4(\mu-\eta:^2\eta^1-OOtBu)_6]$ upon prolonged storage of a toluene solution of the former at ambient temperature, and probe the catalytic activity of the former alkylperoxide in the epoxidation of $trans$-chalcone as a model enone.

## Results and discussion

**Synthesis and structures of magnesium alkylperoxides incorporating a fluorinated β-diketiminate ligand.** The parent $tert$-butylmagnesium complex, $[(^{f5}BDI)MgtBu(Et_2O)]$ (1), was easily prepared in almost quantitative yield by salt metathesis using a solution of $tert$-butylmagnesium chloride in Et$_2$O and a lithium salt of $^{f5}$BDI-H (for $^1$H NMR spectra see Supplementary Fig. 1). The controlled oxygenation of a toluene solution of 1 at −20 °C followed by concentration of the reaction mixture and crystallization at −30 °C led to the reproducible isolation of a dimeric magnesium $tert$-butylperoxide $[(^{f5}BDI)Mg(\mu-\eta:^2\eta^1-OOtBu)]_2$ $2_2$ (we note that the isolated alkylperoxide is not solvated by the Et$_2$O, for $^1$H NMR spectra see Supplementary Fig. 2). In contrast to our previous observation[14], increase in the oxygenation temperature does not influence the reaction outcome, and compound $2_2$ was also the only isolable product from the oxygenation of 1 at room temperature. Thus, these results demonstrate that subtle change in the fluorinated β-diketiminate ligand may govern the stability of magnesium alkylperoxide subunit. Much to our surprise, extended storage of a toluene solution of $2_2$ for ca. 7 days at room temperature followed by concentration of the reaction mixture and crystallization at −30 °C allowed to isolate a minute amount of a spiro-type, tetranuclear magnesium $tert$-butylperoxide $[(^{f5}BDI)_2Mg_4(\mu-\eta:^2\eta^1-OOtBu)_6]$ (3) (Fig. 1b). Based on our current knowledge on the chemistry of magnesium alkylperoxides, the uniqueness of the compound 3 seems manifold. Firstly, the formation of 3 strongly contradicts with the previously encountered ubiquitous transformation of the magnesium alkylperoxides to the respective alkoxides[13–15]. Whilst it is premature to propose a detailed mechanism that accounts for the formation of 3, it seems reasonable to suggest that $2_2$ is kinetically labile in a toluene solution and very slowly undergoes a ligand scrambling[24,47–49]. Moreover, the molecular structure of 3 can formally be considered as the first structurally authenticated homoleptic magnesium $tert$-butylperoxide $[Mg(\mu-OOtBu)_2]_2$ terminated by two monomeric magnesium $tert$-butylperoxides $[(^{f5}BDI)Mg(OOtBu)]$. Finally, a spiro-type structure has hitherto been encountered for zinc alkoxides and is without precedent in the chemistry of non-redox-metal alkylperoxide[47]. Thus, the isolation of magnesium alkylperoxide 3 appears to be a turn-up for the books owing to the exceptional instability of this class of compounds.

The identities of the peroxides $2_2$ and 3 were unambiguously established by single-crystal X-ray diffraction (for details see Supplementary Figs. 3 and 4 and Supplementary Tables 1–4). Compound $2_2$ crystallizes in the triclinic space group P-1 as a dimer with the bridging $tert$-butylperoxide group with $\mu-\eta:^2\eta^1$ binding mode (Fig. 2a). The Mg⋯Mg separation between magnesium centres equals 2.927(1) Å. The Mg–O bond lengths (2.003–2.026 Å.) fall in the typical range observed for other magnesium alkylperoxide systems[13–15]. The O–O bond length (1.485(2) Å) is slightly longer to that found in the previously

**Fig. 1 The oxygenation of alkylmagnesium complexes stabilized by β-diketiminate ligands. a** Previous studies on the reactivity of alkylmagnesium complexes towards $O_2$. **b** The oxygenation of a *tert*-butylmagnesium complex incorporating a β-diketiminate ligand with perfluorinated N-aryl substituents.

reported dimeric (1.442(3) Å and 1.383(6) Å)[14] and monomeric (1.480(5) Å)[15] magnesium alkylperoxides incorporating the β-diketiminate ligands. The differences in the O–O bond length likely result from the presence of the additional electron-withdrawing fluoro substituents in the β-diketiminte ligand, which decrease the electron density on the alkylperoxide moiety. The alkylperoxide **3** crystallizes in the triclinic space group P-1 as a tetranuclear aggregate with an almost linear arrangement (Fig. 2b). Four magnesium centres are bridged by six *tert*-butylperoxide groups with μ-η:2η[1] binding mode (Fig. 2c). The terminal five-coordinate magnesium atoms are capped by the fluorinated β-diketiminate ligands, whereas both inner magnesium centres are six-coordinate. The O–O bond length of 1.386(4) Å in the central bridging *tert*-butylperoxide groups is shorter than the analogous bond distance in the remaining *tert*-butylperoxide ligands (1.482(3) Å in O(1)–O(2) and 1.454(3) Å in O(3)–O(4)).

**Reactivity of the alkylperoxide $2_2$ in model oxygen-transfer reactions.** Having in mind high oxidation propensity of the magnesium alkylperoxides, we wondered if $2_2$ can oxidize the parent *tert*-butylmagnesium complex **1** with the formation of the corresponding magnesium alkoxide, as commonly postulated[13].

To this aim, a toluene solution of **1** was added to a solution of $2_2$ at −20 °C and then the reaction mixture was stirred up for 24 h at ambient temperature (Fig. 3a). Interestingly, the alkylperoxide $2_2$ was inert toward the parent alkylmagnesium compound and the subsequent crystallization from the reaction mixture resulted in full recovery of $2_2$. Thus, our results demonstrate that neither metathesis reaction leading to the respective magnesium alkoxide, nor the ligand redistribution process might be observed under these conditions.

Non-redox-metal alkylperoxides may act as an efficient catalysts in the epoxidation of electron-deficient olefins. However, in contrast to well established Zn-based catalysts[50–54], the application of the magnesium relatives still remains a highly unexplored research area. Up to now, there are two magnesium-based catalytic systems for the epoxidation of enones[55,56], of which only one is well-defined[14]. Moreover, electron-withdrawing β-diketimine ligands with different fluorine-containing substituents may have a profound effect on the activity of various catalytic systems[14,57]. Thus, in the next step, the activity of $2_2$ as a potential catalyst in the epoxidation of electron-deficient olefins with TBHP as an oxidant was tested. In a control experiment, *trans*-chalcone as a model enone, TBHP and catalytic amount of $2_2$ were mixed in toluene at 0 °C and the reaction was conducted further at this temperature (Fig. 3b). The conversion to epoxide

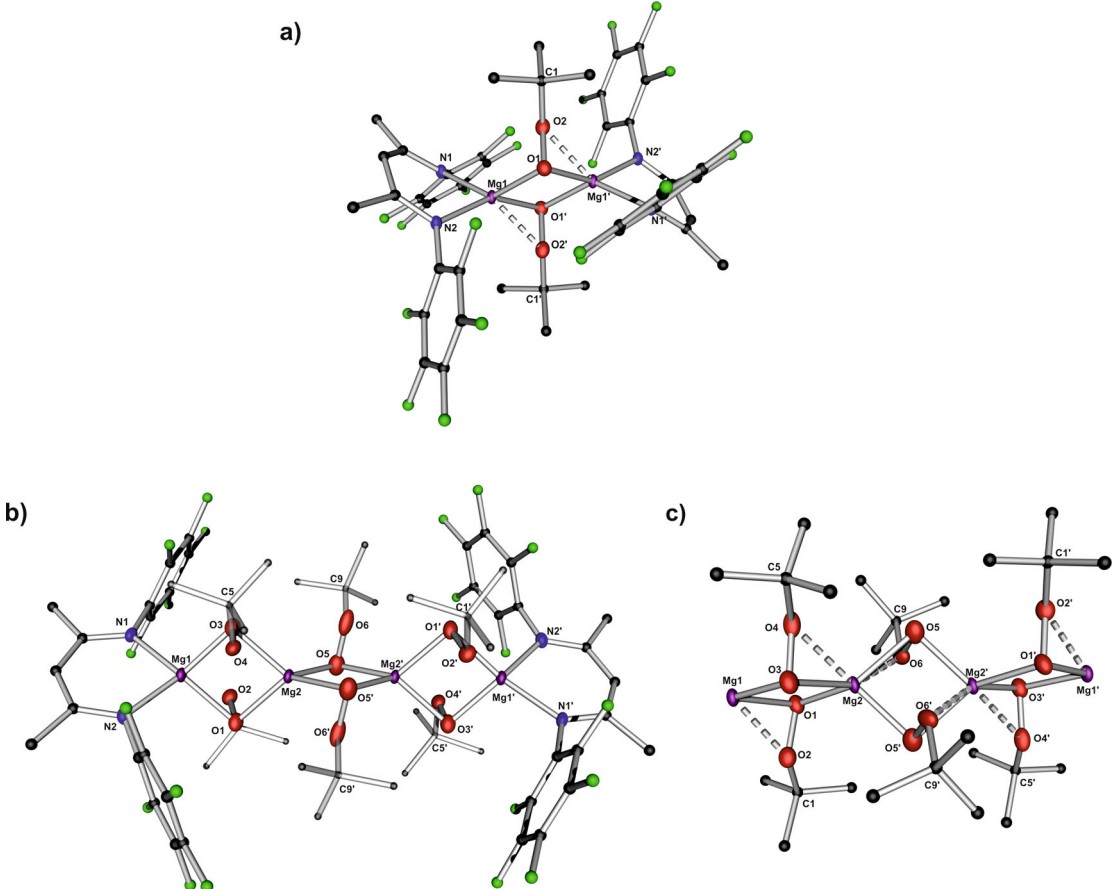

**Fig. 2 Molecular structures of the magnesium alkylperoxides. a** The molecular structure of $2_2$. **b** The molecular structure of 3. **c** An inset of the spiro magnesium alkylperoxide skeleton. Ellipsoids for Mg-, O- and N-atoms are set at 35% probability. Hydrogen atoms are omitted for clarity.

was monitored by gas chromatography. Our results clearly indicate that $2_2$ exhibited high activity in the epoxidation reaction and the 2-benzoyl-3-phenyloxirane was formed in almost quantitative yield after ca. 20 min. Thus, the presence of the perfluorinated N-aryl substituents of the β-diketimine ligand in $2_2$ results in a significant increase in the catalytic ability in comparison to the non-fluorinated [($^{dipp}$BDI)Mg(μ-OO$t$Bu)]$_2$-based catalytic system. The observed reaction time places the investigated system among the most active known catalysts for the epoxidation of enones[1–3]. For example, catalytic epoxidation of *trans*-chalcone using an ill-defined magnesium system $n$Bu$_2$Mg/diethyl tartrate/TBHP or a well-defined zinc system (N,N′)ZnOO$t$Bu/TBHP (where N,N′ = enaminooxazolinate ligand) takes 24 h[58] or 1 h[52], respectively. Thus, the described relatively rare example of structurally well-defined catalysts, its very high activity and the potential to build up a metal-friendly catalytic system for the asymmetric epoxidation of enones appears to be main advantages of these studies. The observed formation of **3** during the prolonged storage of $2_2$ indicates a dynamic behaviour of magnesium alkylperoxide in a toluene solution, which is likely associated with the cleavage of one or both alkylperoxide bridges. The simplified catalytic cycle for the oxidation of *trans*-chalcone is shown in Fig. 3c. On this stage of studies, we may propose that the magnesium alkylperoxide reacts with *trans*-chalcone with the formation of the epoxide and the corresponding *tert*-butylmagnesium alkoxide. Next, the subsequent reaction of the latter with TBHP allows to recover the catalyst and closes the catalytic cycle (Fig. 3c).

## Conclusion

The elucidation of molecular structure and reactivity of non-redox-metal alkylperoxides has posed significant difficulties for decades. In this report we fill a gap in the molecular-level knowledge of magnesium alkylperoxides. The controlled oxygenation of *tert*-butylmagnesium complex supported by the β-diketimine ligand with the perfluorinated N-aryl substituents led to the dimeric magnesium alkylperoxide, which does not transform to the corresponding magnesium alkoxide at ambient temperature. Instead, ligand scrambling process with the formation of a spiro-type, tetranuclear magnesium *tert*-butylperoxide was observed for the first time. The formation of the latter not only demonstrates an exceptional stability of the alkylperoxide unit towards further transformations, but also highlights that there is much to be clarified with respect to the solution behaviour of magnesium alkylperoxides. Interestingly, the magnesium alkylperoxide [($^{f5}$BDI)Mg(μ-η:$^2$η$^1$-OO$t$Bu)]$_2$ bearing the electron-withdrawing fluorine atoms on the N-aryl substituents in the supporting β-diketiminate ligand possesses higher catalytic activity in comparison to the [($^{dipp}$BDI)Mg(μ-OO$t$Bu)]$_2$-based catalytic system with the non-fluorinated N-aryl substituents. Moreover, despite the high oxidation properties of the dimeric *tert*-butylperoxide incorporating the perfluorinated β-diketiminate ligand in the catalytic epoxidation of enones, its σ-bond metathesis reaction with the parent alkyl did not occur. Overall, our studies provided an important aspect of knowledge on the stability of magnesium alkylperoxide compounds. We believe that these intriguing findings will stimulate further

**Fig. 3 Reactivity of the magnesium alkylperoxide 2$_2$ in model oxygen-transfer reactions. a** Reaction of the magnesium alkylperoxide 2$_2$ with the parent *tert*-butylmagnesium complex 1. **b** Catalytic epoxidation of *trans*-chalcone mediated by magnesium alkylperoxide 2$_2$. **c** The simplified catalytic cycle for the oxidation of *trans*-chalcone.

systematic studies on the chemistry of non-redox-metal alkylperoxides as well as support the rational design of $^{NR}$MOOR reagents for various oxygen-transfer organic processes, including efficient catalytic systems for the asymmetric oxidation of electron-deficient olefins.

## Methods

All reactions were conducted under argon atmosphere using standard Schlenk and glovebox techniques (<0.1 ppm $O_2$, <0.1 ppm $H_2O$). Toluene, benzene, *n*-hexane and *n*-pentane were degassed with nitrogen, dried over activated aluminium oxide (MBraun SPS) and stored over 3 Å molecular sieves. Deuterated solvents were dried over Na/K alloy and distilled under argon atmosphere, and a solution of *tert*-butylmagnesium chloride in Et$_2$O (Sigma-Aldrich) was used as receive. 2-[(2,3,4,5,6-Pentafluorophenyl) amino]-4[(2,3,4,5,6-pentafluorophenyl)imino]-pent-2-ene ($^{f5}$BDI-H) was synthesized according to the literature procedure[57]. The oxygenation reactions were carried out using pure dioxygen dried by passing it through a tube filled with anhydrous NaOH/KOH. NMR spectra were acquired on Varian Inova 500 MHz and Varian Mercury 400 MHz spectrometer at 298 K. Elemental analysis were performed using an UNI-CUBE (Elementar Analysensysteme GmbH).

**Caution**. The metal alkylperoxides are potentially explosive and should be handled with care; however, we have not encountered this type of sensitivity in the investigated compounds under the studied conditions.

**Synthesis of [($^{f5}$BDI)MgtBu(Et$_2$O)] (1)**. A yellow solution of $^{f5}$BDI-H (0.538 g, 1.25 mmol) in toluene (10 mL) was cooled to −78 °C and 0.47 mL of a solution of *n*BuLi in heptane (1.31 mmol) was slowly added by means of a syringe. The reaction mixture was allowed to warm to room temperature and stirred for 24 h. During this time, change of the solution colour from yellow to orange crystals was observed. Next, a solution of *t*BuMgCl (1.25 mmol, 1.0 mL) in Et$_2$O was slowly added by means of a syringe at room temperature and the reaction mixture was stirred overnight. Then the volatile materials were removed under reduced pressure and the product was extracted into toluene (10 mL) and filtered. The volatile materials were removed under reduced pressure. The resulting residue was washed

with cold pentane (3 × 5 mL) and dried under vacuum yielding **1** as an oily product with essentially quantitative yield. The product was used in the next step without further purification. $^1$H NMR (C$_6$D$_6$, 400.0 MHz, 298 K): δ [ppm] = 0.94 (t, 6H, Et$_2$O), 1.02 (s, 9H, C(CH$_3$)$_3$), 1.69 (s, 6H, CCH$_3$), 3.56 (q, 4H, Et$_2$O), 4.88 (s, 1H, α-CH) ppm.

**Synthesis of [($^{f5}$BDI)Mg(μ-OOtBu)]$_2$ (2$_2$)**. Method 1: A solution of **1** (233 mg, 0.4 mmol) in toluene (6 mL) was cooled to −20 °C. Next, the Schlenk headspace was replaced by dry $O_2$ for 1 h. The headspace was then replaced by dry $N_2$ and the resulting solution was concentrated. Compound **2$_2$** was obtained as yellow crystals after crystallization at −27 °C (98 mg). Isolated yield 45%. Method 2: **1** was dissolved in toluene under an $N_2$ atmosphere at room temperature. Next, the Schlenk headspace was replaced by dry $O_2$ for 1 h under this conditions. The headspace was then replaced by dry $N_2$ and the resulting solution was concentrated. Compound **2$_2$** was obtained as yellow crystals after crystallization at 0 °C (88 mg). Isolated yield 41%. $^1$H NMR (400 MHz, C$_6$D$_6$, 298 K): δ [ppm] = 1.00 (s, 18H, C(CH$_3$)$_3$), 1.37 (s, 12H, CCH$_3$), 4.84 (s, 2H, α-CH) ppm. $^{13}$C{$^1$H} NMR (125 MHz, C$_6$D$_6$, 298 K): δ [ppm] = 171.68, 129.15, 128.39, 125.22, 98.52, 82.64, 32.48, 25.11, 23.55; elemental analysis: calcd (%) for C$_{42}$H$_{32}$N$_4$O$_4$F$_{20}$Mg$_2$: C 46.48; H 2.97; N 5.16; found C 46.65; H 3.09; N 5.09.

**Synthesis of [($^{f5}$BDI)$_2$Mg$_4$(μ-OOtBu)$_6$]$_2$ (3)**. A yellowish solution of **2$_2$** (100 mg, 0.18 mmol) in toluene (5 mL) was storage at ambient temperature for ca. 1 week under an $N_2$ atmosphere. Next, the resulting solution was filtered and concentrated. A few crystals of compound **3** was obtained after crystallization for 7 days at −27 °C. Elemental analysis: calcd (%) for C$_{58}$H$_{68}$N$_4$O$_{12}$F$_{20}$Mg$_4$: C 46.74; H 4.60; N 3.76; found C 46.81; H 4.68; N 3.71.

**Metathesis reaction of 1 and 2$_2$**. A solution of **1** in toluene was added (0.16 mmol, 93,5 mg, 2 mL) to the stirred solution of **2$_2$** (0.08 mmol, 87 mg, 2 mL) in toluene at −20 °C. The reaction mixture was warmed to room temperature and the reaction was continued for 24 h. Compound **2$_2$** was obtained after concentration of the reaction mixture followed by crystallization overnight at −27 °C.

**Catalytic epoxidation of *trans*-chalcone mediated by 2₂.** To a toluene solution of $2_2$ (0.1 mmol, 109 mg, 4 mL) at 0 °C, the solutions of the chosen enone (1.0 mmol, 0.25 M) and TBHP (2.0 mmol, 4.0 M) in toluene was added. Next, the reaction were continued at this temperature and the conversion was monitored by gas chromatography. In order to prepare the sample, 1 mL of the reaction mixture was hydrolysed with a saturated aqueous solution of KF (2 mL) followed by the drying of the organic phase with anhydrous MgSO₄.

**Crystallographic measurements for 2₂ and 3.** The crystals of all complexes were selected under Paratone-N oil, mounted on the nylon loops and positioned in the cold stream on the diffractometer. The X-ray data for complex $2_2$ and 3 were collected at 100(2)K on a Nonius KappaCCD diffractometer using CuKα radiation (λ = 1.54184 Å) and MoKα radiation (λ = 0.71073 Å), respectively. The data were processed with CrysAlisPro[59]. The structures $2_2$ and 3 were solved by direct methods and refined using the SHELXL97[60]. All non-hydrogen atoms were refined with anisotropic displacement parameters. Hydrogen atoms were added to the structure model at geometrically idealized coordinates and refined as riding atoms.

## Data availability

The authors declare that the data supporting the findings of this study are available within the paper and its supplementary information files. Crystallographic data (excluding structure factors) for the structures $2_2$ and 3 reported in this paper have been deposited at the Cambridge Crystallographic Data Centre under deposition numbers CCDC-($2_2$)-2045430 and CCDC(3)-2045431. These data can be obtained free of charge from The Cambridge Crystallographic Data Centre via www.ccdc.cam.ac.uk/data_request/cif. The crystallographic information of compounds $2_2$ and 3 is available in Supplementary Data 1 and 2, respectively.

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

## Acknowledgements

The authors would like to acknowledge the National Science Centre - Grant OPUS DEC-2017/27/B/ST5/02329, and European Union, Regional Development Fund within a project of Foundation for Polish Science - POMOST/2013–8/15 (K.Z.) and Foundation for Polish Science START scholarship no. 064.2021 (T.P.) for financial support. T.P. is a recipient of a scholarship awarded by the Polish Ministry of Education and Science to outstanding young scientists.

## Author contributions

T.P. carried out the synthetic experiments and analysed the experimental data. K.Z. analysed the experimental data. Z.O. analysed the data concerning the epoxidation reaction. K.N. performed the experiments on the metathesis reaction. I.J. carried out the X-ray single-crystal structure analyses. J.L. originated the central idea, coordinated the work and analysed the experimental data. J.L. and T.P. wrote the manuscript.

## Competing interests

The authors declare no competing interest.
