## [Peer Review File. · Communications Chemistry]

Reviewers' comments:

Reviewer #1 (Remarks to the Author):

Building upon their previous studies, Lewinski and coworkers report the structure and reactivity of new magnesium alkylperoxide complexes stabilized by highly electron-deficient pentafluorophenyl-substituted beta-diketiminate ligands, which were prepared by the controlled oxygenation of the corresponding tert-butylmagnesium complex 1. The dimeric magnesium alkylperoxide $[(f5BDI)Mg(\mu-\eta^2:\eta^1-OOtBu)]_2$ (complex 2) were found to be more stable than the non-fluorinated analogue and show higher catalytic activity in the epoxidation of trans-chalcone using tBuOOH as a terminal oxidant. The use of the f5BDI ligand also allowed the authors to observe further transformations of complex 2 involving ligand scrambling without decomposition to a magnesium alkoxide, to afford a novel spiro-type, tetranuclear magnesium alkylperoxide $[(f5BDI)_2Mg_4(\mu-\eta^2:\eta^1-OOtBu)_6]$ (complex 3), which was structurally characterized by single crystal X-ray diffraction. This complex represents the first non-redox-metal alkylperoxide in a spiro-type structure, and contains a unique homoleptic magnesium tert-butylperoxide $[Mg(\mu-OOtBu)_2]_2$ structure. Overall, this study has revealed unique structures and reactivities of magnesium alkylperoxides and advanced the underexplored chemistry of magnesium alkylperoxides, which is expected to inspire researchers in various fields. Therefore, I recommend the publication of the manuscript in Communications Chemistry after addressing the following issues.

- 1) Because the key enabler in this work is the highly electron-deficient pentafluorophenyl-substituted beta-diketiminate ligand (f5BDI), I recommend the authors should explicitly draw the chemical structures of Ar substituents on BDI ligands in Figure 1a, for readers to easily recognize the structural differences.
- 2) While some reactivities and catalytic activity of complex 2 are examined, no information on those of complex 3 is available. Some discussion should be made. For example, does complex 3 work as an active catalyst for the epoxidation as well?
- 3) To arouse the interest of a broad readership of Commun. Chem. I recommend the authors examine if the epoxidation system with complex 2 is more broadly applicable to other olefins (not limited to enones) since only a single example of trans-chalcone is shown in the current manuscript.
- 4) Is it possible to use O₂ as a terminal oxidant for the epoxidation? If yes, and if it's only possible with the complex 2 stabilized by the f5BDI ligand, it would be interesting.

Reviewer #2 (Remarks to the Author):

In a very nice contribution the Lewiński group reported the synthesis of the dimeric magnesium tert-butylperoxide $[(f5BDI)Mg(\mu-\eta^2:\eta^1-OOtBu)]_2$, first. The new feature of this compound is the fluorinated β -diketiminate ligand. The new complex was well characterized. The title compound slowly decomposes to a very interesting spiro-type, tetranuclear magnesium alkylperoxide. Finally, the reactivity of the title complex for the catalytic epoxidation of trans-chalcone with tert-butylhydroperoxide was investigated.

The described chemistry is nice and the manuscript should be accepted for publication. However, some points should be clarified beforehand:

- NMR data of 3 should be collected.
- Figure 3 bottom shows the reaction of a magnesium alkoxide to the peroxide in the catalytic cycle. It would be nice to see if this reaction works in a preparative scale.
- line 217 and 230. Please use subscript for the formulas.
- line 237: it should be 3 and not 5.
- peroxides are potentially explosive. Please add a statement about potential hazards.

Reviewer #3 (Remarks to the Author):

This paper by Lewiński and coworkers is a nice piece of work. It is well written (with only a few grammatical errors). The primary claim concerns the development of a very reactive epoxidation catalyst involving Mg. The background to this type of reaction is the generally poorly understood nature of the catalytic species involved - here the catalysts is investigated in detail. I agree with the central claims of the paper and the work is important in this area.

I suppose one of the only things that I think is lacking is some sort of substrate scope - there is only one alkene investigated and the relative advantages and disadvantages of this system compared to reported ones is not proved (these other studies do this). A better overall comparison could be made if they were to epoxidize a greater set of substrates and perhaps at least one more challenging alkene. They could argue with some justification, however, that this will be the follow-up paper. A final point concerns the mechanism. They propose a rather simple (but convincing enough one) but it would be nice to see an NMR study of the reaction order (e.g., rate of depletion of the alkene with time) which would back the mechanism up. Again, I would be happy to see this type of analysis in a full paper on this at a later stage.

The work should be published in my opinion.

WARSAW UNIVERSITY OF TECHNOLOGY
FACULTY OF CHEMISTRY

ul. Noakowskiego 3
00-664 Warsaw
Poland
tel: +48(22)234-7315
E-mail: lewin@ch.pw.edu.pl

Professor Janusz Lewiński FRSC

Warsaw, July 3, 2021

Replies to the Reviewer Comments

Reviewer 1

General remark: *Overall, this study has revealed unique structures and reactivities of magnesium alkylperoxides and advanced the underexplored chemistry of magnesium alkylperoxides, which is expected to inspire researchers in various fields. Therefore, I recommend the publication of the manuscript in Communications Chemistry after addressing the following issues*

Reply: We are very happy that Reviewer appreciated our work and found it inspiring for the community. Below, we refer to the Reviewer's specific comments.

Specific comments:

Remark 1: *Because the key enabler in this work is the highly electron-deficient pentafluorophenyl-substituted beta-diketiminate ligand (f5BDI), I recommend the authors should explicitly draw the chemical structures of Ar substituents on BDI ligands in Figure 1a, for readers to easily recognize the structural differences.*

Reply: Indeed, this is a valid point and in the revised version Figure 1a has been changed accordingly.

Remark 2: *While some reactivities and catalytic activity of complex 2 are examined, no information on those of complex 3 is available. Some discussion should be made. For example, does complex 3 work as an active catalyst for the epoxidation as well?*

Reply: As we noted in the main text, the knowledge about magnesium alkylperoxides is in its infancy and only started to gain momentum. Owing to their frequently encountered exceptional reactivity, the controlled synthesis and subsequent isolation of this family of compounds still appear to be a challenge. To continue our systematic investigations on the influence of steric and electronic contributions of a supporting ligand on the reactivity of magnesium alkylperoxides., and in particular searching for an efficient ligand scaffold for magnesium alkylperoxide-based catalysts for the asymmetric epoxidation of enones, in this work, we turned our attention to β -diketimine ligands with fluorinated N-aryl substituents. As far as we are concerned, the isolation of the spiro-type, tetranuclear magnesium alkylperoxide 3 is an seminal albite unexpected

result in particularly for magnesium alkylperoxides. Further investigations on the reactivity of 3 will be very challenging, yet time-consuming, and seem to be out of the scope of our communication. It will probably be many years before other examples of spiro-type metal alkylperoxide emerge which subsequently should pave the way for systematic studies on their reactivities. In turn, one should not expect that this type of metal alkylperoxides can act as a potential catalytic system for the asymmetric epoxidation.

Remark 3: *To arouse the interest of a broad readership of Commun. Chem. I recommend the authors examine if the epoxidation system with complex 2 is more broadly applicable to other olefins (not limited to enones) since only a single example of trans-chalcone is shown in the current manuscript.*

Reply: While this is valid point, this suggestion seems to be out of the scope of our manuscript. The reported results well substantiate and complement challenging studies on the chemistry of magnesium alkylperoxide and we believe that these intriguing findings will stimulate further systematic studies on the rational design of ^{NR}MOOR reagents/catalysts for various oxygen-transfer organic processes, including efficient catalytic systems for the asymmetric epoxidation of electron-deficient olefins. Moreover, one should be aware that such extended and specific investigations on the scope of organic substrates are particularly desired using catalytic systems mediated by metal complexes supported by chiral ligands, and they are out of the scope of our manuscript. As long as there is progress in development of zinc alkylperoxides-based catalytic systems (our ongoing and previous studies indicate that this type of catalysts are generally very efficient for the epoxidation of a broad scope of enones, cf. Adv. Synth. Catal. 2016, 358, 864), our knowledge on a well-defined and effective catalyst based on magnesium alkylperoxides is only fragmentary. Hence, studies in this area will be a subject of another project and published due course.

Remark 4: *Is it possible to use O₂ as a terminal oxidant for the epoxidation? If yes, and if it's only possible with the complex 2 stabilized by the f5BDI ligand, it would be interesting.*

Reply: To the best of our knowledge, compound involving non-redox metal centers are out of scope of catalytic systems using O₂ as a terminal oxidant for the epoxidation, and catalyst based on transition metal centers are commonly used on this occasion. In the case of compound 2, the presence of *tert*-butylhydroperoxide is essential in the catalytic cycle proposed on Figure 3c.

Reviewer 2

General Remark: *In a very nice contribution the Lewiński group reported the synthesis of the dimeric magnesium *tert*-butylperoxide [(f5BDI)Mg(μ-η²:η¹-OOtBu)]₂, first. The new feature of this compound is the fluorinated β-diketiminato ligand. The new complex was well characterized. The title compound slowly*

decomposes to a very interesting spiro-type, tetranuclear magnesium alkylperoxide. Finally, the reactivity of the title complex for the catalytic epoxidation of trans-chalcone with tert-butylhydroperoxide was investigated. The described chemistry is nice and the manuscript should be accepted for publication.

Reply: We are very pleased that the Reviewer appreciate our manuscript. Below, we refer to the Reviewer's specific comments.

Remark 1: *NMR data of 3 should be collected.*

Reply: We want to stress that the isolation of the spiro-type, tetranuclear magnesium alkylperoxide **3** is an seminal albite unexpected result in particularly for magnesium alkylperoxides. An elementary ¹H NMR spectrum for compound **3** was performed, however the spectrum was relatively featureless (to avoid confusion we do not refer to this questionable experiment at this stage of our investigations). Due to very limited amount of compound **3**, more advanced NMR studies could not be perform, and the identity of this intriguing compound was unambiguously established by single crystal X-ray diffraction.

Remark 2: *Figure 3 bottom shows the reaction of a magnesium alkoxide to the peroxide in the catalytic cycle. It would be nice to see if this reaction works in a preparative scale.*

Reply: We agree with the Reviewer's point of view but again we would like to stress that such studies involving magnesium peroxides are very challenging. Therefore, initially such investigations have currently been performed for the relatively easier to control reaction systems based on zinc compounds incorporating fluorinated β -diketiminatate ligands and will be published due course.

Remark 3: *line 217 and 230. Please use subscript for the formulas.*

Reply: In the revised version of our manuscript, the formulas was corrected.

Remark 4: *line 237: it should be 3 and not 5.*

Reply: In the revised version of our manuscript, the text was corrected.

Remark 5: *peroxides are potentially explosive. Please add a statement about potential hazards.*

Reply: In the revised version of our manuscript, the appropriate text was added

Reviewer 3

General Remark: *This paper by Lewiński and coworkers is a nice piece of work. It is well written (with only a few grammatical errors). the primary claim concerns the development of a very reactive epoxidation catalyst involving Mg. The background to this type of reaction is the generally poorly understood nature of*

the catalytic species involved - here the catalysts is investigated in detail. I agree with the central claims of the paper and the work is important in this area.

Reply: We are very happy that Reviewer appreciated our work and found it important in this area. Below, we refer to the Reviewer's specific comments.

Remark 1: *I suppose one of the only things that I think is lacking is some sort of substrate scope - there is only one alkene investigated and the relative advantages and disadvantages of this system compared to reported ones in not proved (these other studies do this). A better overall comparison could be made if they were to epoxidize a greater set of substrates and perhaps at least one more challenging alkene.*

Reply: While this is valid point, this suggestion seems to be out of the scope of our manuscript. As we noted earlier, we are on the stage to search for a relatively stable, well-defined and effective catalyst based on magnesium alkylperoxides. In turn, the reported results well substantiate and complement challenging studies on the chemistry of magnesium alkylperoxide and we believe that these intriguing findings will stimulate further systematic studies on the rational design of ^{NR}MOOR reagents/catalysts for various oxygen-transfer organic processes, including efficient catalytic systems for the asymmetric epoxidation of electron-deficient olefins. At this moment such extended and specific investigations on the scope of organic substrates are extensively studied in our group using catalytic systems mediated by zinc alkylperoxides supported by chiral ligands (for the preliminary studies, see *Adv. Synth. Catal.* 2016, 358, 864)

Remark 2: *A final point concerns the mechanism. They propose a rather simple (but convincing enough one) but it would be nice to see an NMR study of the reaction order (e.g., rate of depletion of the alkene with time) which would back the mechanism up.*

Reply: The stoichiometric reaction between a magnesium alkylperoxide and *trans*-chalcone was performed and the process is essentially diffusion controlled, which exclude the possibility of kinetic studies for this reaction system using standard techniques. In the same time, we would like to mentioned that the respective kinetic investigations have currently been performed for the relatively slower catalytic asymmetric reaction based on zinc alkylperoxides and will be published due course.

REVIEWERS' COMMENTS:

Reviewer #2 (Remarks to the Author):

The authors properly addressed my remarks.

On the other hand, two other reviewers ask for the epoxidation of a few more substrates. I am a bit surprised that not at least one or two more substrates were reacted.

Reviewer #3 (Remarks to the Author):

The authors have done all that is necessary as far as I can see to allow this work to be published in the current form. It will be very interesting to see where this work leads to in terms of substrate scope but I agree that this does not have to be included in this communication. I recommend that the work is accepted.